# Structure, Function and Regulation of the Plasma Membrane Calcium Pump in Health and Disease

**DOI:** 10.3390/ijms23031027

**Published:** 2022-01-18

**Authors:** Joachim Krebs

**Affiliations:** Department of NMR-Based Structural Biology, Max Planck Institute (MPI) for Biophysical Chemistry, 37077 Gottingen, Germany; jkrebs@nmr.mpibpc.mpg.de

**Keywords:** plasma membrane calcium pump, PMCA, calmodulin, spliced isoforms, microdomains

## Abstract

In this review, I summarize the present knowledge of the structural and functional properties of the mammalian plasma membrane calcium pump (PMCA). It is outlined how the cellular expression of the different spliced isoforms of the four genes are regulated under normal and pathological conditions.

## 1. Introduction

Ca^2+^ is an essential signal carrier for life. To fulfill its pivotal role as a second messenger in eukaryotic cells, Ca^2+^ homeostasis is tightly controlled to maintain the free Ca^2+^ concentration in the cell within the range between 100 and 300 nM [1,2,3]. The extracellular calcium level is adjusted in the millimolar range, basically controlled by seven transmembranes spanning the Ca^2+^-sensing receptor, similar to the G-protein-coupled receptor superfamily [4]. This permits Ca^2+^ to function as a versatile extracellular first messenger.

Ca^2+^ can regulate important aspects of cellular activity by participating in many different signal transduction pathways, leading to fertilization to create life and ending life with programmed cell death [5]. Since Ca^2+^ can be a toxic component of the cell by precipitating inorganic phosphate at millimolar levels, it is essential to control intracellular Ca^2+^ in the nanomolar range [6]. Therefore, the maintenance of calcium homeostasis is a highly integrated process controlled by a number of hormonally regulated feedback loops involving an elaborate system of Ca^2+^ transporters, channels, exchangers, Ca^2+^-binding/buffering proteins and Ca^2+^ pumps (see Figure 1). It is of pivotal importance that Ca^2+^ fluxes into and out of the cell are tightly regulated to permit the rapid changes of cellular Ca^2+^ levels, which is a prerequisite for Ca^2+^, enabling its signaling function [7], see Figure 1.

This review focuses on the structure, function and regulation of the plasma membrane calcium pump (PMCA), which is one of a number of membrane Ca^2+^-transporting proteins important for maintaining cellular Ca^2+^ at the low free concentration essential for its signaling function [1].

## 2. General Properties of PMCA

In 1961, Dunham and Glynn published the observation that, in erythrocytes, a Ca^2+^-dependent ATPase should exist [8]. In 1966, Schatzmann provided direct evidence that Ca^2+^ is pumped out of human red blood cells at the expense of ATP against a steep Ca^2+^ gradient across the plasma membrane [9]. In 1977, an important aspect of the regulation of the plasma membrane calcium pump (PMCA) was published by three independent groups [10,11,12]. These authors reported that the PMCA of erythrocytes could be stimulated by a recently discovered Ca^2+^-dependent protein, now known as calmodulin. Since calmodulin interacted directly with PMCA in a Ca^2+^-dependent way [13], this permitted Niggli et al. to isolate PMCA in pure form from erythrocytes using a calmodulin-affinity column [14]. Similar to SERCA, the calcium pump of the sarco/endoplasma reticulum, the sodium/potassium pump of the plasma membrane, or SPCA, the calcium pump of the Golgi apparatus, and PMCA belong to the so-called P-type pump family [15,16]. These enzymes are characterized by forming a high-energy acyl-phosphate during the reaction cycle (Figure 2), in most cases an aspartylphosphate, providing the protein with sufficient energy to pump ions across the membrane against a steep ion gradient [17].

Recently, a cryo-EM structure of PMCA complexed with neuroplastin was reported, but the structure was obtained in the absence of calmodulin [25]. The published structure resembles the structure of the SERCA pump in the E1 conformation and is similar to our modelled E1 structure of PMCA [24], see Figure 3.

As defined by Toyoshima for the SERCA pump, PMCA also displayed different regulatory domains such as the actuator domain (**A**; Figure 3), the nucleotide-binding domain **N** and the phosphorylation domain **P**. In 1988, the primary structure of PMCA was determined by cloning it from humans [26] and rat tissues [27]. As can be noticed from Figure 3, 10 transmembrane helices could be identified, similar to the SERCA pump. The major protein mass is protruding into the cytosol (Figure 4). Later studies identified PMCA as an essential component of all mammalian plasma membranes represented by four different genes (PMCA 1–4), which in humans are localized on four different chromosomes [28,29].

In addition, various spliced isoforms could be identified for the four different genes due to the existence of two splice sites. Site “A” is located within the first intracellular loop, which is part of the actuator domain and is close to the phospholipid-binding site (Figure 4). Site “C”, however, was identified within the calmodulin-binding domain [30], see also Figure 4. These splicing variants could lead to up to 30 different spliced isoforms, with proteins of various molecular weights representing different functional and localization properties, as is discussed later.

The activity of PMCA can be regulated in many ways. Since the enzyme is a membrane protein, the lipid composition of the membrane turns out to be important. Acidic phospholipids such as phosphatidylserine or phosphatidic acid, as well as polyunsaturated fatty acids, but not neutral phospholipids such as phosphatidylethanolamine can stimulate PMCA [18,31,32]. In detailed studies, two putative phospholipid-binding sites were identified, one located in the first cytosolic loop of PMCA [33], see Figure 4, the other one situated within the calmodulin-binding domain, thereby indicating that acidic phospholipids can compete with calmodulin to activate PMCA [34], but detailed structural information is not available to date. It is also noticeable that PMCA can be clustered in ER-PM microdomains [35]. On the other hand, it is important to note that PMCA was associated with caveolae [36,37] or with lipid rafts [38]. In caveolae, PMCA is colocalized with caveolin-1, the major protein of caveolae [39]. The importance of this colocalization between caveolin-1 and PMCA was underlined by the observation that, in mice for whom caveolin-1 was ablated, the function of PMCA4b was impaired [39]. In lipid rafts, which are thought to be related with but not identical to caveolae [40], PMCA is often stimulated by phosphatidyl-4,5-bisphosphate (PIP_2_), a highly conserved regulator of Ca^2+^ transport, even in yeast [41]. PIP_2_ is almost as effective as calmodulin in stimulating the enzyme [42,43]. To understand the complex interaction between PMCA and its various regulators in the presence of different phospholipid environments, detailed structural studies such as cryo-EM would be essential.

As outlined later, an important function of PMCA to control Ca^2+^ homeostasis in subcellular microdomains is its ability to interact with protein complexes via a PDZ recognizing motif in the C-terminal region of PMCA. This is found in all “b” spliced isoforms of the enzyme, but not in the “a” isoform [44]. In this context, an interesting observation was recently reported by Go et al. [45], who demonstrated that POST (partner of STIM), a regulator of store-operating calcium entry [46] (SOCE), preferentially associated with PMCA4b rather than 4a to stimulate PMCA4 function, in order to decrease the Ca^2+^ concentration in the micro-domain near to the plasma membrane [45]. This preference of associating POST with PMCA4b rather than 4a may occur due to the PDZ-domain-recognizing motif, which is present in PMCA4b, but not in 4a. On the other hand, PIP_2_, another regulator of Orai1/STIM1 clustering to enable SOCE may also be important to control PMCA activity essential for Ca^2+^ clearance in the neighborhood of the Orai/STIM complex. This may facilitate SOCE in junctions between the endoplasmic reticulum and the plasma membrane, stabilized by the remodeling of the actin organization network, which can also regulate PMCA [45,47,48,49,50].

The activity of PMCA can increase significantly by oligomerization [51] or by phosphorylation [52]. The most dominant activator of PMCA is calmodulin, which directly interacts with the enzyme using a specific binding domain close to the C-terminus of PMCA [13]. This localization is typical for the animal kingdom, in contrast to PMCA in plants where the calmodulin-binding domain is usually located at the N-terminus [53,54]. 

Most calmodulin-dependent enzymes need an intact calmodulin to be fully activated. In this respect, PMCA is unique since the enzyme can be fully activated by the C-terminal half of calmodulin (but with lower efficiency). By contrast, the N-terminal half of calmodulin is unable to activate PMCA [55] This is further corroborated by using a genetically modified calmodulin, in which each of the four calcium-binding loops of the protein are mutated to prevent calcium binding [56]. By using a calmodulin in which either the two calcium-binding loops of the C-terminal half were mutated, PMCA was hardly activated in contrast to the mutants in the calcium-binding loops of the N-terminal half of calmodulin which were able to fully activate the calcium pump [57]. This special feature of PMCA, for which the C-terminal half of calmodulin is the dominant part for activating the enzyme, was in contrast to other calmodulin-dependent enzymes [57]. This may be the reason that, at resting condition of the cell, PMCA may not be totally in the autoinhibition state. Such an assumption is based on the finding that the peptide C20W, representing the N-terminal part of the calmodulin-binding domain of PMCA binds with a high affinity to calmodulin. This already occurs at 200 nM Ca^2+^ concentration [58], which corresponds to the free Ca^2+^ concentration of a resting cell. From this peculiar property of PMCA one can conclude that, in the absence of calmodulin, PMCA is not in a fully autoinhibited state, as observed for other calmodulin-dependent enzymes [59,60], even if the calmodulin-binding domain of PMCA interacts in the absence of calmodulin with two internal receptor sites of PMCA located within the catalytic part of the pump [61,62], see Figure 4.

As mentioned before, one of the splice sites, site “C”, is located in the middle of the calmodulin-binding domain, giving rise to a number of spliced isoforms (see Figure 4). These spliced isoforms of PMCA contain a high variety of the calmodulin-binding domain which results in altered lengths of the C-terminal amino acid sequence due to a change of the reading frame. As can be noticed from Figure 4, due to splicing in an additional exon, exon 21 for PMCA1, a complex variety of different spliced isoforms occurs. Only the spliced isoform “b” excludes this additional exon and keeps the calmodulin-binding site intact. 

Since PMCA is a ubiquitous enzyme in all mammalian cells and participates in regulating their Ca^2+^ homeostasis [7], the importance of a global regulatory role was overestimated for a long time in comparison to the much more powerful role of the SERCA pump or the sodium/calcium exchanger [1]. However, in recent years, it became more apparent that tuning Ca^2+^ homeostasis in different microdomains of the cell is significantly controlled by PMCA [35]. As already indicated before, this is made possible due to a PDZ-binding motif [44] which is located downstream of the calmodulin-binding domain at the C terminus of the PMCA amino acid sequence (Figure 4). Therefore, PMCA is recruited to target protein complexes containing PDZ domains [44,63,64,65,66,67].

Several attempts were made to obtain structural information on the interaction of calmodulin with its targets. Upon the binding of Ca^2+^, calmodulin changes its conformation to enable it to interact with distinct binding domains of targets. The first structure of calmodulin interacting with a peptide representing the binding domain of myosin light chain kinase was solved either by NMR [68] or by X-ray crystallography [69]. These structures revealed the basic recognition pattern that calmodulin wraps around its target in a Ca^2+^-dependent manner using two hydrophobic anchor residues to change the extended dumbbell shape structure of calmodulin into a more globular structure. By using small-angle X-ray scattering (SAXS), we investigated the interaction of calmodulin with peptides of different lengths corresponding to the calmodulin-binding domain of PMCA [70]. It was interesting to observe that, if the peptide lacked the second hydrophobic anchor residue (C20W), the resulting complex displayed a dumbbell-type structure similar to intact calmodulin. However, if the binding peptide included this second anchor amino acid (C24W), a globular structure of the complex was obtained [70].

To gain more detailed information of the calmodulin–C20W complex, we solved the structure by using NMR [71,72] confirming the extended structure of the complex as predicted by the SAXS measurements (Figure 5a). However, binding calmodulin to C28W, which contains the second anchor residue, permits calmodulin to wrap around the binding domain [73], see Figure 5b). 

Structural insights into the activation process of PMCA by calmodulin were also obtained for the plant PMCA ACA8 of *Arabidopsis thaliana*, which contains the calmodulin-binding region at the N-terminus of the ACA8 amino acid sequence [74]. It was surprising to note that the entire regulatory domain of ACA8, in a complex with Ca^2+^-calmodulin, revealed a 2:1 stoichiometry. The target for calmodulin binding was a long-stretched helix with calmodulin binding to site 1 in an antiparallel orientation, whereas binding to site 2 calmodulin was oriented parallel to the target helix. Biochemical data provided evidence that both calmodulin-binding domains were essential for the autoinhibition of the enzyme indicating a two-step, calmodulin-dependent activation mechanism of ACA8 [74].

## 3. The Different Isoforms of PMCA

In humans, as well as in mammals, PMCA is generally represented by four different genes. PMCA1 and 4 are ubiquitously expressed, and thus are thought to be the “housekeeping forms”. PMCA2 and 3 are highly expressed in excitable cells and are known for their high basal activity, in contrast to PMCA1 and 4 for which the basal activity is much lower. In addition, a plethora of spliced isoforms with different functional and distribution properties were detected [17,30]. Therefore, two splice sites were identified in PMCA of human or murine origin. Site “A” is located closely upstream of the phospholipid-binding site (see Figure 4), whereas site “C” is identified within the calmodulin-binding domain located close to the carboxy terminus of PMCA (Figure 4). As can be noticed from Figure 4, splicing at site “A” gives rise to isoforms “x” or “z” for genes 1, 3 or 4, depending on the size of the exon to be spliced in or to be excluded. In case of PMCA2, the splicing pattern at site “A” is more complex, resulting in isoforms “w”, “x” or “z” due to splicing in or is excluded of nucleotides of different lengths (see Figure 4). It is important to note that the size of the exons to be spliced in at site “A” does not interfere with the reading frame in contrast to splicing at site “C”, for which the reading frame can be either intact (e.g., isoforms “b”, “c” or “d”; Figure 4) or is changed (isoforms “a”), since the number of nucleotides to be spliced in interferes with the reading frame, resulting in a shortened carboxy-terminal sequence. Therefore, splicing at site “C” reveals a calmodulin-binding domain with considerable variety, leading to PMCA isoforms with variable affinities for calmodulin, resulting in different activation profiles of PMCA [17,30,75].

### 3.1. PMCA1

PMCA1 is ubiquitously expressed and, therefore, is thought to be the house-keeping form of the plasma membrane calcium pump. This is in contrast to PMCA2 and 3, which are expressed only in specialized tissues, as discussed later. The dominant role of PMCA1 is also documented by the finding that PMCA1 knockout mice are embryonically lethal in contrast to PMCA4 [76,77]. As mentioned before, different spliced isoforms can be detected for PMCA1. Of special interest is the distribution pattern of PMCA1a and b in the developing rat brain as first described by Brandt and Neve [78]. These authors detected PMCA1b at E10, the earliest embryonic day studied, but PMCA1a was only faintly visible. By comparing the mRNA levels of PMCA1b and 1a, it was surprising to realize that the level of mRNA of PMCA1b declined during brain development in contrast to mRNA 1a; the expression of the latter constantly increased. This peculiar observation of changing the expression between the two PMCA1 isoforms a and b was later confirmed on the protein level by using isoform-specific antibodies [79]. These observations during rat brain development may suggest that PMCA1a was specifically important for the maturation process of neuronal development, an interpretation which was later confirmed by Kip et al. for hippocampal neurons [80]. By using immunohistochemistry, Kenyon et al. demonstrated a detailed pattern of the distribution of PMCA1a in neuronal plasma membranes of stomata, dendrites and spines [81]. An important consequence comparing PMCA1a with 1b is the difference in the interaction of the calmodulin-binding domain with the internal receptor sites of PMCA, as identified by Falchetto et al. [61,62]. Due to splicing in of exon 21, containing 154 nucleotides (see Figure 4), into the calmodulin-binding domain of PMCA1a, the resulting change of the reading frame is due to an early stop codon which results in a shorter C-terminal amino acid sequence of the calmodulin-binding domain, and subsequently, is translated into a shorter amino acid sequence of the carboxy terminus of PMCA1a. This likely implies that the interaction with the second receptor site within the second cytosolic loop of the PMCA [62] results in a reduced efficiency of this interaction with the consequence of a reduced autoinhibition of the enzyme.

On the other hand, this change in the structural properties of PMCA1a compared to 1b, with an intact calmodulin-binding site and high sensitivity towards calmodulin activation, provides an advantage of PMCA1a in fast-spiking neurons. Such a possible difference in the activation properties between those two isoforms could be comparable to the well-studied difference in the calmodulin-dependent activation kinetics between PMCA4a and 4b as studied by Caride et al. [82]. These authors reported a higher efficiency of PMCA4a as compared to PMCA4b to reduce cytosolic Ca^2+^ concentration after a Ca^2+^ spike. How the alternative splicing of PMCA1 to obtain the 1a isoform is possibly regulated by the thyroid hormone-induced CaMKIV was extensively described recently [17,83,84].

Korthals et al. investigated the influence of different PMCA isoforms on the development of mice B-cells, since appropriate Ca^2+^ signaling is crucial for the development of those cells [85]. It was surprising to observe that, in mice where PMCA4 was ablated, B-cells developed normally in contrast to mice where PMCA1 was conditionally knocked out. Those mice presented a significant reduction in B-cells of different origins. Those cells presented elevated basal levels of Ca^2+^ concentration and a reduced Ca^2+^ clearance. From this report, it can be concluded that the presence of a specific isoform of PMCA can be important for the appropriate development of certain cell types [86].

PMCA1 is the dominant isoform of the intestine to control calcium homeostasis [87]. The expression of PMCA1 in the intestine is regulated by Vitamin D [88]. Since mice for which PMCA1 are deleted in the germ line are embryonically lethal (PMCA1^−/−^; [77]), Ryan et al. [89] selectively ablated PMCA1 in the intestine of mice (Pmca1^EKO^ mice) by using the cell-specific gene inactivation Cre-lox technique [90]. The Pmca1^EKO^ mice are viable but are smaller than their wild-type littermates, and, importantly, demonstrate a lack of response to vitamin D_3_ and a decreased bone mineral density [89].

### 3.2. PMCA2

In contrast to PMCA1 and 4 as the basic housekeeping isoforms of mammalian plasma membranes, PMCA2 and 3 are specifically expressed in highly specialized excitable cells, such as the outer hair cells of the inner ear and specific neurons or epithelia of the mammary gland [91]. Brandt et al. studied the developing nervous system of the rat and discovered the isoform PMCA2 [92], which was later confirmed by Hilfiker et al. who characterized the human homologue of PMCA2 and described the presence of spliced isoforms of PMCA2 [93]. PMCA2 turned out to be especially present in Purkinje and in granular cells of the cerebellum [93]. Zacharias and Kappen studied the developmental expression of the four PMCAs in the mouse [94]. They detected PMCA2 as early as 12.5 dpc (days post coitum), restricted to the nervous system, especially the cerebellum [94]. 

As can be noticed from Figure 4, PMCA2 can produce a complex composition of spliced isoforms which determine their tissue distribution and membrane targeting. In a distinguished study, Chicka and Strehler provided convincing evidence that splicing at site “A” within the first intracellular loop of PMCA2 to obtain PMCA2w, which contains an additional insertion of 45 amino acids, was specifically located in the apical membrane of polarized epithelial MDCK cells [95,96]. This was in contrast to the isoform PMCA2b, which localized to the basolateral membrane of the same cell. The authors also demonstrated that the targeting of PMCA2 to the apical membrane was solely dependent on the presence of the “w” spliced isoform and independent of the splicing at site “C” of the carboxyterminus of PMCA (see Figure 4). PMCA2w/a plays a critical role in regulating Ca^2+^ homeostasis in the hair cells of the inner ear; therefore, missense mutations can cause deafness and a loss of balance in mice and humans [97,98]. The pivotal role of expressing PMCA2 in the stereocilia of the inner hair cells was underlined by several mutations of PMCA2, first described in mice, and later confirmed in humans [66], see also Table 1. 

In a recent study, Vicario et al. described a missense mutation of PMCA2 (V1143F) affecting the calmodulin-binding domain of the enzyme interfering with the activation by calmodulin [99]. Molecular dynamic simulations suggested that calmodulin binding to its binding domain of the PMCA2 V1143F mutant was compromised [99]. This resulted in an impaired PMCA2 not able to participate in clearing Ca^2+^ transients efficiently to keep the basal Ca^2+^ concentration of cerebellar Purkinje neurons under control, which may be especially effective in sub-plasma microdomains. This missense mutation (V1143F) of PMCA2 was detected in a patient [99]. The authors suggest this mutation may be linked to the cerebellar ataxia as diagnosed for the patient.

As mentioned before, PMCA2 is also of special importance for lactating mammary tissue. Reinhardt and Horst reported that the expression of PMCA2w/b during the developing lactation process could be up to 100-fold increased in the apical membrane of mammary epithelial cells [100]. On the other hand, the milk of mice lacking PMCA2w/a had a reduced calcium content by 60% [101]. In breast cancer, the expression of PMCA2 is significantly increased and colocalizes with the human epidermal growth factor receptor 2 (HER2) in actin-rich membrane microdomains [102] (Table 1). By investigating several patients, Romero-Lorca et al. reported the increased expression level of various spliced isoforms of PMCA2 mRNA in breast cancer tumor cells, which correlated with the expression of different growth factor receptors [103].

### 3.3. PMCA3

The genetic isoform PMCA3, which in humans is located on the X chromosome [29], is mainly expressed in presynaptic terminals of the cerebellum, the choroid plexus or the hippocampus [104,105]. In addition, PMCA3 can also be detected in skeletal muscles [106]. In an Italian family, a mutation of PMCA3 (G1107D) was detected, which affected the region of the calmodulin-binding domain and was associated with X-linked congenital cerebral ataxia [107,108]. Based on the NMR structure of calmodulin bound to the peptide, representing the calmodulin-binding domain of the pump (see Figure 5b), it can be concluded that replacing Gly1107 with an aspartate would negatively influence the interaction of calmodulin with its binding domain. This mutation affected not only the activation of the pump by calmodulin, but the autoinhibition of the enzyme was also impaired due to the reduced interaction with the internal receptor sites. Extensive clinical studies of patients with aldosterone-producing adenomas (APA) leading to secondary hypertension were reported (Table 1). Several deletion mutants of PMCA3 were detected within the fourth transmembrane domain of the pump, affecting the efficiency of Ca^2+^ clearance [109,110].

### 3.4. PMCA4

As mentioned before, PMCA4 is ubiquitously expressed in mammalian plasma membranes and, therefore, is next to PMCA1, thought to be a housekeeping form of the pump. In contrast to PMCA1, which if knocked out in mice is embryonically lethal, PMCA4^−/−^ mice appeared to be normal, but PMCA4^−/−^ male mice were infertile [77,111] (Table 1). This different importance between PMCA1 and PMCA4 for mammalian development suggests that PMCA1 is an essential housekeeping isoform, but not PMCA4, since the loss of PMCA4 can be compensated by PMCA1, but PMCA4 cannot overcome the loss of PMCA1. 

An important property of PMCA is the possibility of interacting with other proteins, especially in microdomains, to gain the spatial control of intracellular Ca^2+^ concentrations. It was previously pointed out that, at the C-terminus of PMCA, a special motif is located, which can interact with other proteins via their PDZ domain to build multifunctional protein clusters, as first described by Kim et al. [44]. For this interaction, it is important that the C-terminal sequence of PMCA is intact. This is the case for all “b” spliced isoforms of PMCA (see Figure 4). Kim et al. provided evidence that PMCA4b could interact with the PDZ domain of the membrane-associated guanylate kinase (MAGUK) of COS-7 cells [44]. This led the group of Neyses to the important finding that the signaling of the neuronal nitric oxide synthase (nNOS) in the heart is regulated by PMCA4b via a PDZ-domain-mediated interaction [64]. This was first demonstrated in HEK293 cells [63] and later by generating transgenic mice that overexpressed PMCA4b [64]. These authors also generated mice that overexpressed a modified PMCA4b, lacking the last 120 amino acids of PMCA4b (PMCA ct120), including the PDZ-recognizing motif and the calmodulin-binding domain. PMCA ct120 is a very active pump [112] but it cannot interact with nNOS or modulate its function [64]. The same group later confirmed that the PMCA4b-nNOS complex regulates cardiac contractility in mice via the modulation of cyclic nucleotide signaling in compartmentalized microdomains at the plasma membrane [65].

A R268Q missense mutation in PMCA4 was detected in a Chinese family with autosomal, dominant, familial, spastic paraplegia [113,114] (Table 1). This mutation is located close to splice site A and the phospholipid-binding domain (see Figure 4), which may lead to a partly misfolded protein responsible for the dysregulation of Ca^2+^ signaling causing neuronal deficits associated with FSP.

Naffa et al. made the interesting observation that PMCA4b can inhibit the migration of BRAF-mutant melanoma cells and impair their metastatic activity [115]. The authors suggest that by remodeling the actin cytoskeleton, PMCA4b affects the cell migration of those cells. The authors further report that the remodeling of the actin cytoskeleton by PMCA4b induces a severe change in the cell shape of the melanoma cells resulting in their polarization with the consequence of a reduced aggressive melanoma phenotype. The authors conclude that PMCA4b could have a general role in influencing the cell shape, an observation also made in breast cancer cells.

## 4. Concluding Remarks

The proteins of the mammalian plasma membrane Ca^2+^ pump (PMCA) are translated from four different genes. Their numerous spliced isoforms differ in their tissue distribution and their regulation of expression. The structural properties of the pump and its regulation were discussed. Numerous mutations of different PMCA isoforms were described and the possible pathological consequences were outlined. A general role of PMCA in controlling Ca^2+^ homeostasis in the cell was, for a long time, overestimated, but recently it became more evident that the PMCA was important for the fine tuning of Ca^2+^ concentration in special microdomains. As outlined in detail before, an important example of such a microdomain is the endoplasmic reticulum-plasma membrane (ER-PM) junctions, which are associated with store-operating calcium entry (SOCE). Since PIP_2_ and POST are sensitive regulators of PMCA, as well as of SOCE [42,45]; thus, PMCA may be associated with SOCE to fine tune local Ca^2+^ concentration within the ER-PM junction [45,48,50,116]. This interesting new aspect of regulating Ca^2+^ concentration by PMCA in cellular microdomains will certainly be investigated in more detail in the future.

## Figures and Tables

**Figure 1 ijms-23-01027-f001:**
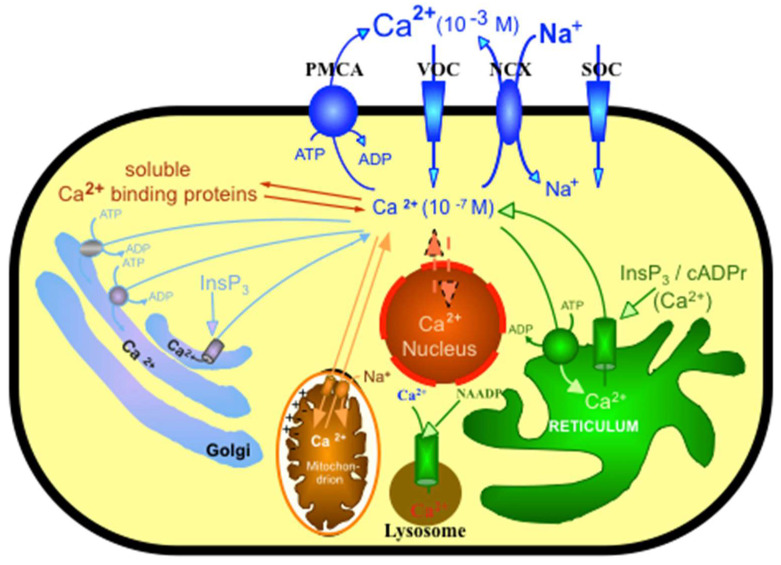
Schematic representation of some of the toolkits controlling Ca^2+^ homeostasis of the cell. The plasma membrane Ca^2+^ pump (PMCA), a Ca^2+^ entry channel, such as VOC (voltage operating calcium channel) or SOC (store operating calcium channel), and the Na^+^/Ca^2+^ exchanger (NCX) are located in the plasma membrane. Intracellular organelles such as the mitochondrion, the sarco/endoplasmic reticulum or the golgi apparatus sequester Ca^2+^ reversibly. This figure was adopted from Krebs et al. *Helv. Chim. Acta*, 2003, *86*, 3875–3888, and reproduced with the permission of the publisher.

**Figure 2 ijms-23-01027-f002:**
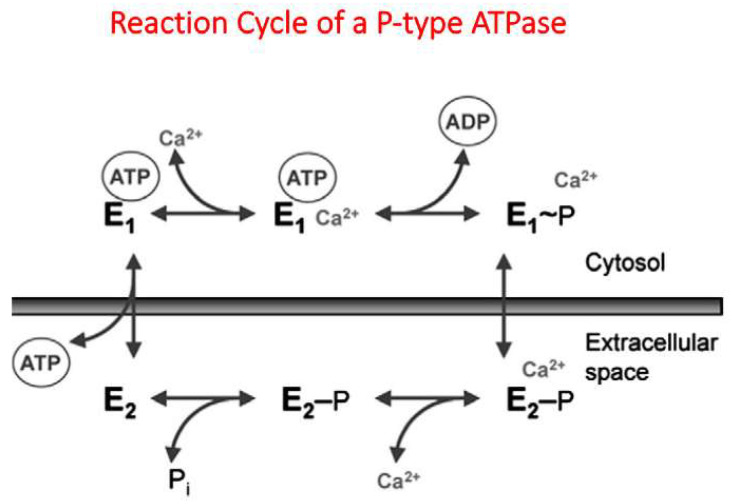
In the case of PMCA this results in a 1:1 ratio of Ca^2+^/ATP [18], in contrast to the SERCA pump with a 2:1 ratio [19]. During the reaction cycle of the PMCA, one could distinguish at least two different conformational states, E1 and E2 [20], see also Figure 2. As later reported by Toyoshima et al. for the homologous SERCA pump, more different conformational states could be observed. They solved the high-resolution structure of the SERCA pump at different stages of the reaction cycle using X-ray crystallography [21,22,23]. Since the structural changes of PMCA during the reaction cycle should be similar to those of the related SERCA pump, homologous modeling for the E1 to E2 transition was attempted based on the structures of the SERCA pump [24], see Figure 3.

**Figure 3 ijms-23-01027-f003:**
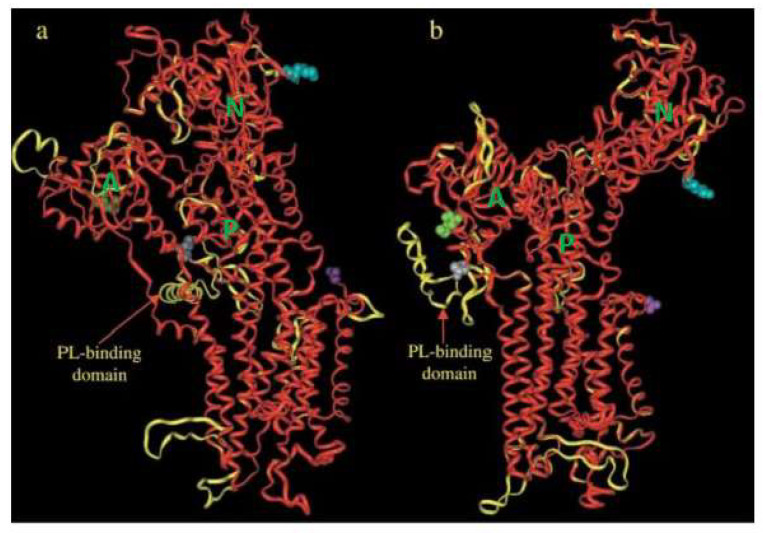
Homology modeling of the PMCA pump based on structures of the SERCA pump, (**a**) represents the Ca^2+^-free (E2) and (**b**) the Ca^2+^-bound (E1) forms. A = actuator domain, N = nucleotide-binding domain, P = phosphorylation domain. The two structures are shown as an overlay of the backbones in ribbon representation, SERCA pump in red, PMCA pump in yellow. There are 37 amino acids of the N-terminal and 144 amino acids of the C-terminal (containing the calmodulin-binding domain) of the PMCA pump, which are missing since they have no correspondence in the SERCA pump. The figure was adopted from Krebs et al. *Helv. Chim. Acta*, 2003, *86*, 3875–3888, and reproduced with the permission of the publisher. For more details on how the homology modeling was obtained [24].

**Figure 4 ijms-23-01027-f004:**
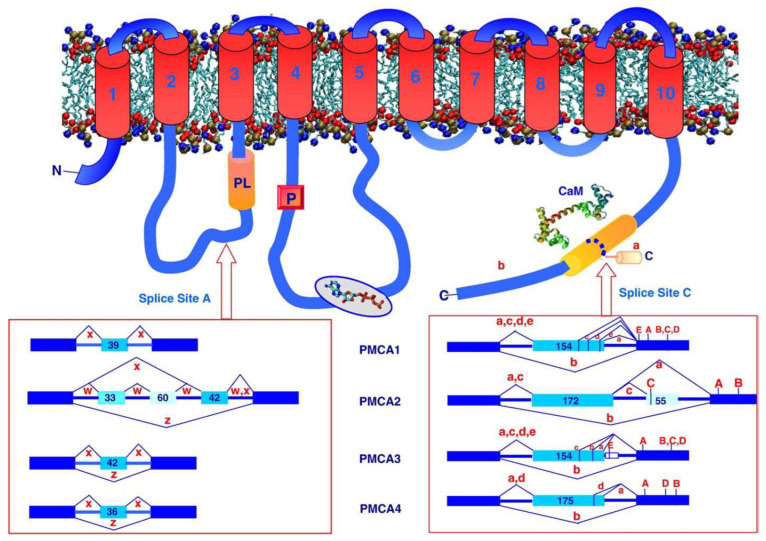
Schematic topology model of the plasma membrane calcium pump (PMCA) and the splicing variants of the four different mammalian genes. Red arrows indicate splice cite “A” within the first cytosolic loop and splice cite “C” within the calmodulin-binding domain at the carboxy terminal of PMCA. The different exon structures affected by alternative splicing are indicated for the 4 different genes of PMCA. CaM = calmodulin, indicated by its structural model in the extended form; P = location of the aspartyl-phosphate intermediate; PL = phospholipid-binding domain. This figure was taken from Krebs, *Biochem. Biophys. Acta* 2015, *1853*, 2018–2024, reproduced with the permission of the publisher.

**Figure 5 ijms-23-01027-f005:**
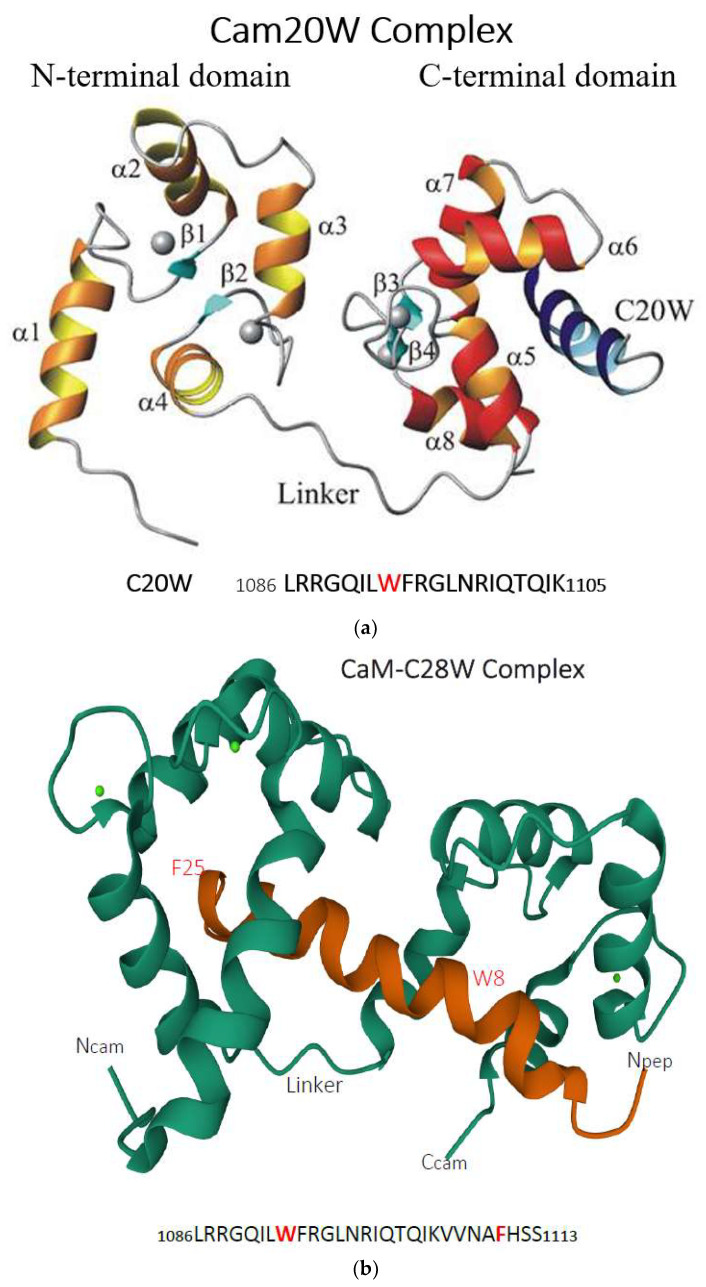
(**a**) Ribbon presentation of the calmodulin/C20W complex in the presence of calcium. C20W corresponds to the N-terminal part of the calmodulin-binding domain of PMCA (PMCA1b 1086–1105), lacking the second hydrophobic anchor as it is present in C28W (see (**b**)). The figure was taken from the PhD thesis of Dr. Bettina Elshorst (2000) and reproduced with the kind permission of the author. (**b**) Ribbon presentation (calmodulin in green, C28W in red) of the calmodulin–C28W complex in the presence of calcium. C28W corresponds to the amino acid sequence of the calmodulin-binding domain of PMCA4b containing the two anchor residues, W8 and F25. W8 is anchored to the C-terminal part of calmodulin, whereas F25 binds to its N-terminal part. Ncam = N-terminal of calmodulin; Ccam = C-terminal of calmodulin; Npep = N-terminal of the peptide C28W; the C-terminal of the peptide is not visible. The structure was obtained from the protein data bank (2KNE) and was published by [73].

**Table 1 ijms-23-01027-t001:** Summary of different diseases linked to several PMCA isoforms.

PMCA	Location	Destination	KO	Mutation	Disease
1	Ubiquitous		Lethal		
2	Neural	Postsynaptic			
2w		Apical			
2w/a		Stereocilia		G283S	Deafness
				G293S	Unbalanced
2w/b		Lact. gland			
2b		Breast tumor			HER2-positive
2	CaM-BD			V1143F	Ataxia
3	Neural, skeletal mtheuscle	Presynaptic			
3	CaM-BD			G1107D	Ataxia
3	Adenomas			Somatic Mutations	APA
4	Ubiquitous		Male infertility		
4b				R268Q	FSP

CaM-BD = CaM-binding domain; APA = aldosterone-producing adenomas; FSP = Familial spastic paraplegia.

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
