# Peer review of "Structure, Function and Regulation of the Plasma Membrane Calcium Pump in Health and Disease"

_ijms, 2022, doi:10.3390/ijms23031027_

Round 1

Reviewer 1 Report

The review is providing an up-to-date data concerning structure, various isoforms and splice variants of PMCA. This is completed by some data regarding general PMCA function and also differential functions of PMCA isoforms. The author also brings some data from the literature concerning regulation of PMCA, especially through calmodulin. The aspect concerning regulation of PMCA through recruitment in membrane micro-domains and also protein complexes containing PDZ domains is very interesting. However I have the feeling that this aspect could be more developed. I would also advice to have a more specific part, recapitulating the involvement of PMCA and PMCA isoforms in physiopathology together with a table summarizing the principal observations

Author Response

I would like to thank the reviewer for the positive response. The request to develop the ability of PMCA to interact with other proteins via its PDZ recognising motif have been outlined on p. 8 of the revised manuscript (written with red colour). The involvement of PMCA isoforms of the 4 different genes have been outlined and summarised in Table 1. I hope I have detailed sufficiently the requests raised by the reviewer.

Reviewer 2 Report

The author has submitted a manuscript of illustrating a current overview regarding possible important effects of plasma membrane calcium pump on physiological functions which are primarily affected by calcium concentrations. The author searched some eligible literature, resulting in reliable conclusions and perspectives. This issue is of interest, and impact of the overview is strong. My overall concern with the overview describing the current available data regarding molecular basis of function of plasma membrane calcium pump and the information provided may offer something substantial that helps advance our understanding of novel perspective against plasma membrane calcium pump in combination with pathological status in human diseases related to calcium homeostasis in subcellular levels.

The authors are strongly recommended to add a short description regarding the way how calcium concentrations are effectively regulated by using compounds which could selectively bind to plasma membrane calcium pump, which will help the author’s perspective.

Author Response

I would like to thank the reviewer for the positive response. The request to develop the ability of PMCA to control Ca2+ concentrations effectively using specific compounds which selectively bind to PMCA are unfortunately not available like thapsigargin which selectively binds and inhibits the SERCA pump. On the other hand, I have outlined in more detail that PMCA is not specifically involved in controlling cellular Ca2+ concentrations in general, but rather fine tuning Ca2+ in specific microdomains.  I hope I have detailed sufficiently the requests raised by the reviewer.

Reviewer 3 Report

This manuscript reviews structural and functional properties of plasma membrane calcium pump (PMCA). This review provides useful information for understanding  the mechanism and regulation of PMCA in  transporting of Ca2+ in signaling function. This review further highlights the regulation of cellular expression of its four isoforms (PMCA1-4) under pathogenical conditions. I think this review covers the key aspects of structural and functional relationship of PMCA in the presence of several diseases causing mutations. I think this review will be very helpful to understand the importance of PMCA to control calcium ion homeostasis in the cell in health science. This will be very interesting to the readers of International Journal of Molecular Sciences. This manuscript is very  well written. I have some suggestions to improve the impact of the paper.

1) PMCA is a membrane protein and the conformational dynamics of the protein in the presence of lipid bilayer environment is very important to understand  its function and regulation. I think it would be good to include additional studies of PMCA (such as autoinhibition/activation) in the presence of lipid membrane environment.

2) I would suggest author to provide some discussion on the challenges and future perspective of biophysical studies of PMCA in health science.

3) There are some typos. For example: page 2, line 42,  "....  exist. 1966 Schatzmann...." should be " ....  exist. In 1966 Schatzmann....". Manuscript needs proof read.

Author Response

I would like to thank the reviewer for the positive response. As a request by the reviewer I dwelled on the function of PMCA in different lipid environments (p. 7 of the revised manuscript, written in red colour). I further made a point what would be essential to get more detailed structural information of PMCA in the presence of different regulators and/or various lipid environment. The revised manuscript was carefully read to avoid typos and so I hope to have satisfied the requests raised by the reviewer.

Round 2

Reviewer 2 Report

The authors have done a good job responding to reviewer comments and concerns in their revision. I believe the manuscript is significantly improved as a result. Now I recommend that this revised version of the manuscript can be accepted for publication in IJMS journal.